# Immunological Aspects of Human Papilloma Virus-Related Cancers Always Says, “I Am like a Box of Complexity, You Never Know What You Are Gonna Get”

**DOI:** 10.3390/vaccines10050731

**Published:** 2022-05-06

**Authors:** Ehsan Soleymaninejadian, Paola Zelini, Irene Cassaniti, Fausto Baldanti, Mattia Dominoni, Andrea Gritti, Barbara Gardella

**Affiliations:** 1Department of Clinical, Surgical, Diagnostics and Pediatric Sciences, University of Pavia, 27100 Pavia, Italy; ehsan.soleymaninejad01@universitadipavia.it (E.S.); i.cassaniti@smatteo.pv.it (I.C.); fausto.baldanti@unipv.it (F.B.); m.dominoni@smatteo.pv.it (M.D.); andeagritti1205@gmail.com (A.G.); b.gardella@smatteo.pv.it (B.G.); 2Department of Obstetrics and Gynecology, Fondazione IRCCS Policlinico San Matteo, University of Pavia, 27100 Pavia, Italy; 3Molecular Virology Unit, Microbiology and Virology Department, Fondazione IRCCS Policlinico San Matteo, 27100 Pavia, Italy

**Keywords:** human papilloma virus, tumor, humoral immunity, cellular immunity, immunotherapy

## Abstract

The human papillomavirus (HPV) can cause different cancers in both men and women. The virus interferes with functions of the cervix, vulva, vagina, anus in the anogenital area, breast, and head and neck cancer due to the local lesions. The tumors lead to death if not treated as a result of distant metastasis to internal organs and brain. Moreover, HPV attenuates the immune system during chronic infection and releases viral antigens into the tumor microenvironment. The tumors know how difficult is to win the battle with a strong united army of immune cells that are equipped with cytokines and enzymes. They confuse the immune cells with secreting viral antigens. The immune system is equipped with cytokines, a complement system, antibodies, and other secretory proteins to overcome the foreign invaders and viral antigens. However, the majority of the time, tumors win the battle without having all the equipment of the immune cells. Thus, in this review, we describe the recent progression in cellular and humoral immunity studies during the progression of HPV-related cancers. First of all, we describe the role of B, plasmoid cells, and B regulatory cells (Breg) in their functions in the tumor microenvironment. Then, different subtypes of T cells such as T CD8, CD4, T regulatory (Treg) cells were studied in recently published papers. Furthermore, NK cells and their role in tumor progression and prevention were studied. Finally, we indicate the breakthroughs in immunotherapy techniques for HPV-related cancers.

## 1. Introduction

Human papillomavirus (HPV) is the most common sexually transmitted infection (STI) in the world. Most people with HPV do not know they have the infection. They never develop symptoms or health problems from it [1]. While it is diagnosed, through tests or symptoms, the patients face treatment and psycho-social challenges. The negative psycho-social effects of HPV interfere with follow-up analysis and treatment of the disease. The Psycho-Estampa scale is employed for measuring the psycho-social effects of HPV. An increase in the Psycho-Estampa scale is highly related to the augmentation of abnormal cells and cancer-related HPVs. The loss to follow-up was high in the population with a higher score on Psycho-Estampa scale [2]. HPV clades infect the cutaneous and mucosal epithelium as cervical and other anogenital mucosae and the replication cycle is intimately linked to epithelial differentiation.

Worldwide, the risk of being infected at least once in a lifetime among both men and women is 50% [3]. HPV infection, regardless of the HPV type, persists for a long time, and the chronic lesions cause cancer in the end [4]. Globally, cervical cancer is the most common of the HPV-related cancers and accounts for 84% of the cases [5]. However, in the United States and other high-income countries, the number of HPV-related head and neck squamous cell carcinoma (HNSCC) is rising relative to HPV-related cervical cancer [6]. In addition, HPV is related to the occurrence of anogenital cancers such as vulval, vaginal, and penile cancers [7]. Moreover, it seems there is relation between HPV infection and breast cancer [8]. The prevalence of HPV in breast cancer patients is 4–86% percent [9]. The prevalence of HPV-16 and 18 is more than 50% of the breast cancer patients in Iranian women shows [10].

An immune response to the HPV infection at the right time and location can overcome the post-infection problems, while impaired or overactive immune response leads to severe pathological problems [11]. In addition, HPV-related lesions and cancers are increased in immunocompromised patients. About 30% of the women who received the immunosuppressive medications and those who suffer from autoimmune diseases are detected with high risk HPVs, especially HPV-16 [12]. For example, applying gut-selected immunosuppression medication such as Vedolizumab over a number of years in the treatment of Crohn’s disease favors HPV replication. HPV-16 replication in a woman with Crohn’s disease led to severe vulvar intraepithelial neoplasia. Examining the same patient for a year shows occurrence of interepithelial neoplasia in new locations and cervix [13]. Moreover, people diagnosed with the human immunodeficiency virus (HIV) are highly susceptible to HPV-related cervical cancers. The risk ratio of cervical cancer in HIV-infected patients is more than six. In 2018, around six percent of patients diagnosed with cervical cancer were positive for HIV [14].

Thus, in this review, we tried to summarize recent publications related to the immunology of HPV. Additionally, probable immunotherapy techniques that can improve the overall health of HPV-related cancers are investigated.

### 1.1. Human Papillomavirus Structure

Human papillomavirus (HPV) belongs to the Papillomaviridae family. This family has double-stranded, closed, circular DNA genome of approximately 8 kb and a nonenveloped icosahedral capsid [15]. The HPV genome is characterized by 8 protein-coding genes and organized into three functional regions: the upstream regulatory region (URR), early (E) and late (L) region.

The URR, also called the long control region (LCR) is located between the L1 and E6 open reading frames (ORFs) and contains the early promoter and regulatory element involved in viral DNA replication and transcription. The early region, which encodes the E1, E2, E4, E5, E6, and E7 proteins, is involved in viral gene expression, replication or survival and the late region, which code for the viral envelope and for the proteins of the capsid’s structure (L1 and L2). LCR methylation plays a key role in HPV gene expression. Cervical samples analysis of HPV-16 patients with cervical cancer, interepithelial neoplasia and asymptomatic patients reveals the importance of the LCR methylation on HPV-16 genes expression. The methylation that occurs on promoter and enhancer areas of HPV-16 genes have a clinical and pathological values. LCR methylation happens in cervical carcinoma and asymptomatic patients in more than 80% and 70%, respectively. The percentage is about 40% in patients with interepithelial neoplasia. Additionally, HPV-16 LCR CpG islands are mostly methylated on gene promoters [16]. The same methylation phenomenon is seen in CpG islands of early promoters and L1 genes of HPV-16-related anal cancer [17]. Thus methylation can be used as a biomarker for HPV-16 cancer progression. The major viral oncoproteins are characterized by E5, E6, and E7 and contribute to cancer initiation and progression by altering cell cycle regulation [15]. Expression of proteins E6 and E7 is associated particularly with integration of viral DNA into the host genome, malignant transformation, and ultimately progression to cancer [18]. The E6/E7 oncoproteins are the primary transforming viral proteins, and they play an important role in immune evasion by targeting cytokine expression to alter cell proliferation and interferon responses [19].

HPVs can be classified into 5 genera (α, β, γ, μ, and ν) [20] and over 200 types have been identified [21,22,23]. Of these, the largest group is the α group and is characterized by 64 HPVs, that mainly infect mucosal epithelia [24]. A subgroup of 17 mucosal HPV (16, 18, 23, 31, 33, 35, 39, 45, 51, 52, 53, 56, 58, 66, 68, 73, and 82) are referred to as high-risk (HR) HPV types. The 10 other HPV types (40, 42, 43, 44, 53, 54, 61, 72, 73, and 81), are classified as low-risk (LR) or non-oncogenic and cause benign lesions [24]. The HR HPV types are the etiological agents of several cancers, such as those of the cervix, vagina, vulva, anus, penis, and a subset of head and neck cancers (HNCs), particularly oropharyngeal cancers cancer [25,26]. The LR HPV types are associated with anogenital warts (AGWs), some type of cutaneous warts [27] and recurrent respiratory papillomatosis (RRP) [28].

The next largest group is the β group HPVs that mainly infect cutaneous epithelia and 50 types have been identified and characterized. HPVs of the remaining three groups (γ, μ, and ν) normally cause only benign disease [23].

### 1.2. HPV Life Cycle

HPV can infect cutaneous epithelial cells or mucosal tissues and, depending on their tropism, HPV are categorized either as cutaneous or mucosal [29]. The viral particle reaches epithelial cells via micro-wounds in the tissue and via interaction with cell surface receptors, such as integrin α6, which is present in basal cells and epithelial stem cells [30]. The protein L1 binds cellular receptors and undergoes to structural modifications, required for endocytosis of the virion [31,32]. During HPV transition along the endosomal pathway, L1 dissociates from the viral genome and L2 mediates viral egressing from the endosomes, guiding HPV vesicles along microtubules into the nucleus. Upon nuclear entry into the dividing cells of the basal layer, viral early transcription is initiated with the expression of the early proteins E1 and E2 [33].

The productive life cycle of HPV can be grouped into three phases: establishment, maintenance replication, and vegetative or productive amplification [34].

The establishment phase consists of the viral transcription and genome amplification following nuclear entry and it remains episomal in the host cells for a long period of time. In this phase the early viral proteins E1, E2 and E4 increases viral genome replication, while E6 and E7 promote host cell proliferation and prevent apoptosis, since E2 decreases the expression of E6 and E7. Loss of E2 repression function leads to deregulation of viral E6 and E7 oncogenes [35].

After the initial establishment phase, the maintenance phase is initiated. This phase consists of creating a constant number of viral genomes and establishing a chronic infection. Finally, the last step involves vegetative or productive viral replication, with the subsequent production of progeny virions [36,37]. Here, the oncoproteins E6 and E7 expressed at relatively low levels in differentiated cells play a key role by inactivating tumor suppressor proteins (e.g., p53, retinoblastoma protein (pRb)) and activating signal transduction, to ensure that the infected cells remain active and progress to the S phase.

Moreover, vegetative amplification, besides being associated with an increase in HPV genome copy numbers, is also followed by the expression of the structural proteins L1 and L2 (Figure 1) [38].

### 1.3. HPV Associated to Cancers

HPV can infect cutaneous epithelial cells or mucosal tissues, as shown in Table 1.

Cutaneous β-HPV infection occurs in young children through skin-to-skin contact and. β-HPV types, which commonly cause asymptomatic infections at cutaneous sites, can sometimes cause debilitating papillomatosis with associated cancer risk [39], although the role of cutaneous β-HPV types in the development of cancer has been less clear [40].

Mucosal are associated with α-HPV types and infections occur during the first sexual exposures in early adulthood, although non-sexual infection may also be possible [41,42]. α-HPV types causes several forms of anogenital cancers aside from cervical cancer including vulvar, penile, and anal cancers [43]. However, the population level impact of these cancers is small when compared to cervical cancer [43]. Besides cervical cancer, HR-HPVs play a key role during cervical intraepithelial neoplasia (CIN) initiation and progression [44,45]. The most cervical HPV infections (>90%) are resolved by the host immune system within 1–2 years without a chronic infection [46]. However, a minority of HPV infections become persistent and the risk of developing epithelial cell abnormalities or cancer is then increased. In particularly, the persistent and chronic infection of HR-HPV, especially type 16, has been confirmed as the principal risk factor for the initiation and development of squamous cell carcinoma [47].

### 1.4. HPV and T Cells

T cells play a critical role in HPV-related cancers. T cells are observed in about 80% of samples taken from HPV-16-related cervical cancers [48]. In addition, an immunogennetic study of 119 HPV-positive head and neck squamous cell carcinoma (HNSCC) patients divulge the importance of T cells infiltration in the tumor microenvironment [49]. A case–control study on Egyptian women shows the importance of the CD4 T and CD8 T cells in predicting the progression of HPV-related cancers such as breast, and head–neck carcinoma [50]. Even lack of co-stimulatory receptors such as CD28 on T cells makes the HPV more aggressive. Lack of CD28 in the population of T cells causes tree-man syndrome in humans. This syndrome happens due to infection with HPV-2 and HPV-4. In the population with normal CD28^+^ T cells frequency, HPV-2- and HPV-4-related warts cannot survive. Despite the temporary existence of HPV-2- and HPV-4-related warts, skin lesions and warts are seen frequently in CD28^--^ T cells patients [51].

The subpopulations of T cells infiltered into cutaneous and mucosal tissues vary based on the HPV-infected organs. In cervical cancer patients infected with HPV-16, T CD8 cells are predominant. In spite of CD8 T cells, the frequency of CD4 T cells is decreased [48]. There are three subsets of T CD8 cells that are active during head–neck HPV-related cancers. All the subsets express programmed cell death protein-1 (PD-1). The first group of T CD8 cells is like oligopotent stem cells. These cells express PD-1/TCF/ CD45RO and proliferate effectively as are encountered with E2, E5, E6 antigens of HPV on major histocompatibility complex-1 (MHC-1). The second subset has a flexible character, flip flop between stem cell and functional characters. The last group is fully functional and could not proliferate in vitro in presence of the HPV antigens [52]. A subset of CD8 T cells that express CD 103 resides in the tumor microenvironment of HPV-related Oropharyngeal squamous cell carcinoma (OSCC). This subpopulation of CD8 T cells is crucial for the eradication of cancer cells and patient surveillance [53]. Furthermore, recently characterized stem-like memory CD8 T cells (Tscm) can remove tumors induced by HPV-16 in vivo and in vitro. The Tscm response to the tumors is more robust while CD40L is activated [54].

It seems in the samples that are taken from the epithelial layer, the CD8 T cells population is dominant. In spite of epithelial layer, in the stromal layer frequency of CD4 T cells is higher [55]. While T cells are infiltrated to the infected area, there is a cross-talk between HPV-infected tissues, the T cells, and other resident immune cells. In the majority of cases, these cell-cell interactions allow the tumor survival. A systematic review and meta-analysis conducted by Litwin et al. [56] showed that as the HPV-infected tissue progresses toward cancer, the number of T cell subtypes decreases in the tissue. Moreover, the dominant population of T cells infiltrated into the tissues is T regulatory (Treg) cells. Treg cells affect the HPV infection by suppressing the CD8 T cells. An increase in the population of Treg is seen in the most common HPV family such as HPV-16, HPV-18 [50]. Cross-talk between Tim+Treg cells and Galectin-9+monocyte in the HPV-related cervical cancer stimulates the secretion of IL-10 and TGF-β, whereas it prevents IFN and IL-12 genes expression. These phenomena help the tumor to be more aggressive and the mortality rate increases. Interestingly, after removing the tumor, Tim+T cells were decreased [57]. As HPV-related tumors progress, Tregs provide a microenvironment that prevents immune responses. An overall view of HPV-related cancer and T cells response is depicted in Figure 1. Overcoming the effects of Tregs in both systemic circulation and tumor microenvironment during HPV infection is crucial [58]. There are some strategies to eliminate the effects of Tregs in the tumor microenvironment, T-win technology, using Tregs cell receptor inhibitors or immune checkpoints, and photodynamic therapy [58]. Moreover, the majority of CD8 T cells in the infected area are exhausted. Manipulating the T cells with PD-1 blocker and Indoleamine 2,3-dioxygenase (IDO-1) inhibitor, cytotoxicity of CD8 T cells increased [49]. Moreover, peripheral T cell response for HPV-related antigens needs further analysis. A cohort study with oral squamous cell carcinoma (OSCC) patients revealed a marked T central memory response against HPV16 L1, and E6 proteins. On the other hand, E7-specific T-central memory response was deficient. The study suggested that overexpression of PD-L1, induced by HPV16 E7, may be responsible for lymphocyte dysfunction. The weak peripheral T cells’ activity against E7 enhances tumor escape mechanisms [59]. Therefore, further investigation of the effects of E7/PDL-1 on peripheral T cells functionality allows for the design of a new immunotherapeutic approach.

In addition to the conventional T cells, gamma delta T cells (γδ T cells), especially a rare population that produce the IL-17 play a key role in the progression of HPV-related breast cancer. This population is a metastasis promoter with producing a high amount of IL-17 [60]. T cells’ functionality during HPV-related cancer is illustrated in Figure 2.

### 1.5. B Cells and HPV

Humoral immune responses play a pivotal role in cancer patients’ survival especially HPV-related cancers [61]. For instance, tumor-infiltrating B cells in head and neck squamous cell carcinoma (HNSCC) that has been caused by HPV improve the overall health of these patients in comparison to those who are not infected with HPV [61]. Transcriptomic study of HPV-positive and HPV-negative HNSCC samples was taken from gene expression omnibus (GEO) and the cancer genome atlas (TCGA) disclosed the importance of B cells especially memory B cells and plasma cells in the life expectancy of the HNSCC patients [61]. Even proteins such as immunoglobulin J polypeptide (IGJ) that are expressed in plasma cells can be used as a predicting factor for the mortality rate of HPV-positive HNSCC patients. The concentration of this protein in the microenvironment is related to the robustness of immune responses to the tumors. There is a relation between mortality rate increase and IGJ decrease [62].

In the HPV-related HNSCC patients, tertiary lymphoid structures (TLS) are seen with mature characteristics. Germinal centers (GCs) are one of the main criteria that show a TLS is mature. In the HPV-related HNSCC patients, B cells are infiltrated into the TLSs with distinctive GCs. These B cells are IgD negative and CD38/BCL6 positive. Moreover, they express the genes that are related to the dark and light zone in common secondary lymphoid organs such as CXCR4 and CD86. Furthermore, they express Semaphorin 4A (SEMA4A) in the GCs of TLSs. GCs with this gene signature demonstrates to help the immune system win the tug-of-war game with the tumor [63]. Effective B cells that impact on overall survival of HNSCC patients are seen in the mice model. B-cells produce a higher amount of IgM and IgG (Figure 3a) when they were reinvigorated with PD-1 blockade and radiotherapy. A decrease in the complementarity-determining region 3 (CDR-3) and an increase in the germinal center (GC) increased the murine survival rate. In addition, increasing the B cells colonies with somatic hypermutations improved the life expectancy of the murine [64].

Immune cells extracted from tumors such as head and neck cancer shows high frequency of B cells in the tumor microenvironment. These cells can produce IgG against HPV in vivo. Titration of IgG is correlated with these B cells. These B cells specifically produce antibodies against HPV antigens, especially E2, E6, and E7. Titration of IgG reveals the most abundant antibody against HPV antigen belongs to E2 (Figure 3b). It sounds that B cells are not the only cells that infiltrate into the tumor microenvironment. Plasma cells and antibody-producing cells are found in high frequency in the tumor milieu. In vitro analysis shows that these cells can produce a high quantity of monoclonal antibodies against HPV. Therefore, in situ plasma cells, B cells, and HPV-specific antibody-producing cells can be employed to produce monoclonal antibodies for vaccination purposes [65]. In addition to the titration of IgG, cytokines such as IL-10 and IL-2 play a crucial role in HPV-related cervical lesions. A matched case–control study in patients with a lower frequency of lesions in their squamous intraepithelial tissue and a higher frequency of lesions in the same tissue disseminates the fluctuation of IL-10 and IL-2 between the cases and control groups. Serum analysis of case and control groups using ELISA shows the probability of the cytokines level in the severity of the lesion [66]. Seroconversion of HPV is related to the reconstruction of the humeral immunity of the patients [66].

Immunohistochemistry and immunofluorescence analysis of HNSCC show the infiltrated cells are mostly CD19/ CD38-positive plasma cells and membrane-spanning 4 domain A (MS4A1)/CD 27-positive memory B cells. Interestingly, recruitment of these subsets of B cells was correlated with the production of B lymphocyte chemoattractant (BLC) or CXCL13 by CD4 T cells [61]. Furthermore, increasing the number of CD20 B cells in the HPV-related oropharyngeal squamous cell carcinoma (OPSCC) induct more CD8 cytotoxic T cells to the tumor area. The tumor-infiltrated B cells secrets a high amount of CXCL9 that commits the T CD8 cells to the tumor milieu. Interaction of these two populations of lymphocytes improves the life span of the OPSCC patients [67]. Moreover, flow cytometry analysis of the peripheral blood mononuclear cells (PBMC), tumor microenvironment, and mucosal area of HNSCC divulges key roles of CD86+/CD21- antigen-presenting B cells and memory B cells with IgD-negative and CD27-positive phenotype in survivance of HPV-related HNSCC patients [68]. In contrast to the positive effects of memory B cells and plasma cells in controlling the HPV-related tumor progression, there is a subset of B regulatory cells (Breg) that plays a contradictive role in the tumor microenvironment. The main markers of Breg cells are CD19, CD1, CD5 that have negative effects on the cytotoxicity of T CD8 cells. A case–control study that compares the ratio of Breg cells in cervical cancer patients and healthy people reveals the increasing number of Breg cells and IL-10. This increase in the number of Breg cells affects the metastasis of the tumor to nearby lymph nodes and tissues, HPV infection, and consequently international federation gynecology and obstetrics (FIGO) stages. IL-10 production by these kinds of cells suppresses the production of granzyme and perforin production of CD8 T cells [69]. However, the activation of a different subset of B cells determines the effectiveness of immune responses toward cancer, the gene expression and epigenetic must be considered as the main factor. It seems methylation of DNA at CpG islands of B cells and other lymphocytes such as T cells improves the overall survival rate of the patients [70].

### 1.6. NK Cells and HPV

The pivotal role of natural killer (NK) cells in HPV was recently reported in patients with normal levels of all types of immune cells, with the exception of mature NK cells. PBMC analysis of HPV-infected patient with skin lesions in a case report study shows an increase in CD8 T cells and NK-like T cells, and a moderate decrease in CD4 T cells. Although the total number of NK cells was normal but effective mature NK cells decreased in comparison to the immature ones. Mature NK cells are positive for CD3, CD56dim, CD16, and KIR, while immature NK cells express CD3, CD56bright, and CD16. A shift in the populations of NK cells, sometimes known as innate lymphoid cell-1, is depicted in Figure 4a. The lack of proper population of NK cells the patient has gone through several surgeries and part of the nose was removed as a result of sever skin lesions. Interestingly, allogeneic transplanting of hematopoietic stem cells restores the functionality of NK cells. Consequently, HPV-related hyperplastic and dysplastic lesions were resigned [71].

It seems HPV, especially the two most common ones, HPV-16 and HPV-18, hamper the functionality of the NK cells. In a follow-up study in women infected with HPV-16 and HPV-18, the in vitro cytotoxicity of the NK cells was analyzed. HPV-16 does not affect the total number of NK cells in the tumor microenvironment. However, NK cells are impaired by expressing muted key proteins such as IL-2 and killer cell lectin-like receptor G1 (KLRG-1). In a follow up study, the NK cells produced enough amount of IFN-γ; however, due to the impairment of IL-2 and KLRG-1 genes expression, the NK cells showed lower cytotoxicity in vitro [72]. In some samples, even IFN-γ production is down-regulated by soluble oncoproteins such as E7 and E6. These two soluble HPV antigens attach to the IL-18 consequently the IL-18\IL-18 receptor binding is impaired. IL-18\IL-18 binding is crucial for IFN-γ production. Moreover, interferon gamma-inducible protein 16 (IFI16) is degraded by soluble E7 antigen and as a result IFN-γ decreases in the tumor microenvironment. Furthermore, production of a regulatory cytokine such as IL-10 and TGF-β induce MHC class I chain related-proteins A (*MICA*) and *B* (*MICB*), CD95 and inhibitory enzymes such as Indolamine-2,3-dioxygenase 1 (IDO), by tumor and regulatory cells suppress the activity of NK cells [73].

It seems the malfunctioning of NK cells in the HPV-infected tumor microenvironment is due to their shifts into intraepithelial innate lymphoid cells-1 (ieILC-1). During HPV-related HNSCC progression, circulating NK cells enter into the tumor microenvironment and as a result of transforming growth factor-B (TGF-β) and IL-15 mostly transform into ieILC-1. The majority of NK cells that transform into ieILC-1 are immature NK cells, as the number of them is high at the tumor site. The ieILC-1 has two subsets, CD49a+CD103+ that prevents tumor growth, and CD49a-CD103+NR4A2 that favors tumor growth. The recent subset of ieILC-1 recruits more Treg cells to the tumor microenvironment and hampers the CD8 T cells activity [74]. Thus, immature NK cells that enter into the tumor microenvironment have three fates; converting into CD49a+ ieILC-1 or CD49- ieILC-1, and exhausted NK cells (Figure 4b). The CD49+a destroys the tumor cells through producing granzyme and perforin. In contrast, CD49a- and exhausted NK cells favor the tumor growth [75]. Moreover, in HPV-16-positive cervical cancer, NK cells activity was hampered due to the downregulation of cell receptors such as NKG2D, NKp46, and NKp30. In this case, intraepithelial lesions increase, and cancer progress [76]. Thus, according to the importance of NK cells in the tumor microenvironment and tumor evasion, they are considered to be one of the valid candidates for HPV-related cancers. Autologous or allogeneic NK cells can be harnessed through gene manipulation, cytokine treatment for HPV-positive cancer immunotherapy [73].

### 1.7. Immunotherapy of HPV-Related Tumors

Impairment of the immune system as a result of chronic infection of HPV notifies the importance of immunotherapy for the immune system homeostasis dysregulation [77]. Clinical trials have documented the effect of immunotherapy in patients with HPV-positive tumors. In a series of clinical trials known as KEYNOTEs (KEYNOTE-012, -40, -37, -048, -55), patients with HPV-related tumors responded well to immunotherapy [78]. A systematic review and meta-analysis that included eleven studies showed better overall survival in patients treated with immunotherapy. The risk ratio in the groups that were administered with immunotherapy was higher. Furthermore, side effects in the groups that received immunotherapy were lower [79]. Another systematic review and meta-analysis with a sample size of 813 patients in seven included studies reassures immunotherapy for HPV-related tumors. Both odds ratio (OR) and overall survival were higher in the HNSCC patients that received PD1/PDL1 blocking antibodies [80]. The clinical trials and systematic reviews studies demonstrate the direction of future HPV-related cancer treatment and encourage immunotherapy methods in this case.

Plasmid-based immunotherapy of HPV-16/18 that targets the E6/E7 antigens, commercially named MEDI0457, increased the number of CD8 T cells in comparison to the Treg cells in HNSCC patients. In this clinical trial study with about twenty patients in the advanced phase of HNSCC, the activity of T cells was analyzed indirectly. IFN-g production was examined through enzyme-linked immune absorbent spot (ELISpot), which is the indirect analysis of T cells activity. However, all the patients responded positively to the treatment, but the majority of CD8 T cells infiltrated in the tumor site were programmed cell death-1 (PD-1)-positive. The author concluded that may a combination of MEDI0457 with anti-PD-1 therapy is a promising way for HPV immunotherapy [81]. There is controversy regarding the efficacy of anti-PD-1 therapy in HPV-related tumors, especially HNSCC. A study with a sample size of more than a thousand patients with HNSCC shows no significant alteration happened in the progression of HPV with people who received or did not receive the anti-PD-1 therapy. Even the overall survival (OS) and the tumor objective rate ratio (ORR) were higher in the patients that express PD-1 [82]. Anti-PD-1 therapy, commercially known as nivolumab, effectiveness for HPV-related HNSCC is improved if combined with other therapeutic techniques. A cocktail of nivolumab, tumor excision, radiotherapy, chemotherapy, and anti-cytotoxic T lymphocyte-associated protein-4 (anti-CTLA4) is more effective than monotherapies. This immunotherapy adjuvant can improve overall survival if applied before tumor surgery. Nivolumab can decrease the tumor stage and after the tumor removal, the overall survival of the patients improves especially HPV+ HNSCC patients [83]. HPV induces immune-checkpoint receptors such as CTLA-4 in the tumor microenvironment. This strategy decreases the functionality of tumor-infiltrated lymphocytes such as cytotoxic T cells. HPV E7 antigen prevents the intracellular expression of Jumonji C histone demethylase 1B (JHDM1B) in the epithelial cells. Next, the CTLA-4 promoter is methylated through H3K36. The methylated promoter is a signal for more transcription and expression of CTLA-4 [84]. So, using ipilimumab in this event can overcome the effects of CTLA-4 in the tumor microenvironments. It has already been seen in the mice model how effective is the combination of anti-PD1/CTLA-4 in overall survivance. HPV+ oropharyngeal tumor mice model treated with antibodies against these two immune checkpoint receptors reveals more than a 90% survival rate. It seems that the treatment increases the frequency of CD8 T cells in the tumor microenvironment. In contrast, the number of Treg and suppressive myeloid cells were downregulated [85].

In addition to blocking antibodies, applying immune cells or genetically modified immune cells for HPV-related tumors has been considered. T cells and chimeric antigen receptor T cells (CAR-T cells) are one of the major subsets of immune cells that are considered to be effective for HPV+ tumors. Employing CAR-T cells for HPV-related epithelial cancer shows some promising signs for further research in this area. In this clinical trial, T cells were engineered to recognize the HPV-E6 antigen. Although the sample size is small, two patients show regression after the treatment. CAR-T cells destroy the metastatic lung tumors in one of the patients. It seems the tumor microenvironment is complicated and genetic differences among the patients affect the immunotherapy process. Two of the patients became resistant to CAR-T cell therapy due to mutation in HLA-A and domains of T cell receptors, respectively. Immune checkpoints such as PD-1 probably play a critical role in moderating the effect of CAR-T cell immunotherapy [86]. It seems CAR-T cells optimization in the case of binding to the HPV-16-related antigens and functionality are the two most important criteria in the next-generation cell therapy procedures. Though genetically engineered T cell receptors (GE-TCR) express better sensitivity toward HPV immunotherapy, by balancing the functionality and avidity of CAR-T cells better results can be reached [87]. The fundamental criterion for an antibody that is used for the backbone of the CAR-T cells is its affinity and binding efficacy. T Cell Receptors (TCRs) are specific without alteration. A clinical trial in which engineered TCRs were used against HPV-E7 antigen shows satisfactory results. However, the sample size was small, 12 patients, but half of them respond very well to the engineered TCRs. Tumors with larger diameters were regressed and smaller tumors disappear in the patients. The major hurdle of the study was a mutation in HLA-A, other genes that were involved in antigen presentation, and the genes that were participated in IFN production. Patients with these mutations did not respond to the treatment properly [88,89]. To overcome this problem, artificial antigen-presenting cells such as modified red blood cells that are known as RTX-321 can be utilized. These engineered red blood cells express HLA-A bonded to E7, tumor necrosis factor ligand superfamily member 9 (4-1BBL), and IL-12 on their membrane. RTX-321 mimics the role of the antigen-presenting cells for CD8 T cells but the results reveal their promising role in activating other immune cells such as NK cells. In addition, titration of IFN-g, tumor necrosis factor-alpha (TNF-a), and CXCL10 was upregulated after coculture of PBMC with RTX-321 [90].

Another important aspect of HPV-related cancer is shifting in the commensal bacteria of tissue such as the vagina and oropharynx and their effects on the innate immunity of the patients. There is a controversy whether *Atopobium vaginae* is part of vaginal microbiota or bacterial pathogenesis. The bacterial pathogenesis hampers the immune responses and dysregulates the proinflammatory cytokines in situ [91]. Furthermore, the abundance of Mycoplasma in the microbial flora of the cervix upregulates the lesion in the cervix of patients that were positive for HIV and HPV [92]. Commensal bacteria reduction, dysbacteriosis, upregulates the titration of inflammatory and wound healing cytokines such as IL-16, IL-1, and TNFα in the HPV-infected lesions. As a result, cells suffer from the disability of genetic materials and DNA damage. Furthermore, Rb and P53 decrease in the tumor microenvironment. HPV becomes more aggressive in this condition and infect to more cells [93]. Moreover, HPV infection disables the NF-KB and Wnt antiviral signaling pathways that help the HPV evasion. Additionally, the disruption of the signaling pathway reduces the essential amino acids for *Lactobacillus* strains’ survivance. *Lactobacillus* strains play an essential role in human vaginal flora and a decrease in their numbers leads to bacterial vaginosis. Oxidative stresses increase due to bacterial vaginosis and it motivates the HPV to replicate faster [94]. Furthermore, commensal bacteria improve the overall responses of immunotherapy procedures. For example, the efficacy of anti-PD1 decreases if bacterial flora of the gut is disturbed in melanoma patients [95].

## 2. Conclusions

HPV-related cancer immunology is receiving more attention due to the immunosuppressive activity of the virus. The humoral immunity consists of B, and plasmoid cells are the main players against HPV-related cancers. These cells producing the IgG and IgM against the HPV antigens such as E6, E7, and E2 neutralize the viral activity. In contrast, B reg cells attenuate the effect of CD8 T cells in the tumor microenvironment. Using monoclonal antibodies such as Ipilimumab and Nivolumab are the most available treatments for HPV-related cancer. In addition, almost all aspects of cellular immunity against HPV-related cancers are suppressed by viral antigens. Tumor antigens such as E6 and E7 or released MICA/B reduce the functionality of immune cells. Furthermore, Treg and Breg cells produce cytokines such IL-10 and TGF-B that mitigate cellular immunity and help the tumor progress. Recent developments in engineering T cells such as CAR-T cells can overcome the attenuated cellular immunity in the tumor microenvironment. The number of Treg cells is high in the tumor milieu. So, engineering the T cells should not be limited to one subset of T cells. For example, Treg engineering can increase the efficacy of this event. Moreover, using the potential artificial antigen-presenting cells such as RTX-321 can be an overturn in cell therapy methods. RTX-321 improves both cellular and humoral immunity against tumors. Utilizing monotherapy may prevent the progression of HPV-related tumors partially. However, the future of HPV-related immunotherapy deals with the immune system from multiple aspects. Thus, we think a combination of engineered bacteria or oncolytic viruses with immunotherapy can revolutionize the immunotherapy of HPV-related cancers. An overview of new techniques that may improve the overall health of HPV-related cancer patients is depicted in Figure 5.

## Figures and Tables

**Figure 1 vaccines-10-00731-f001:**
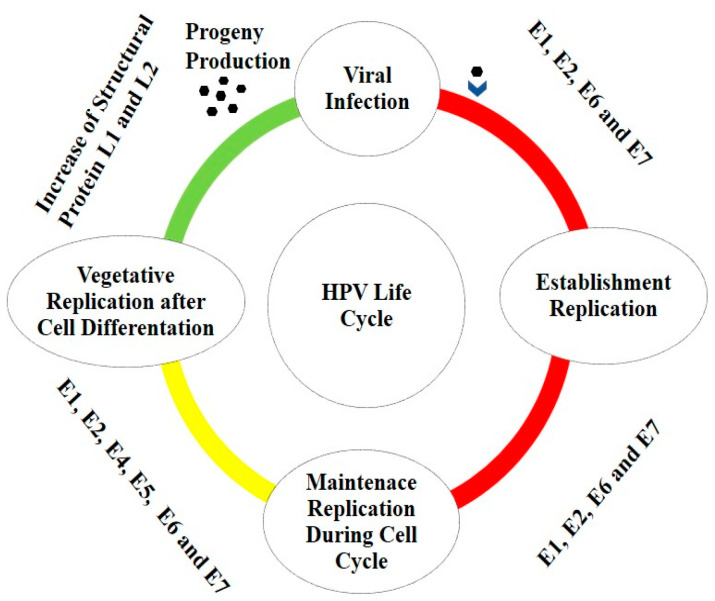
HPV life cycle. The first step of the replication cycle of HPV, called establishment replication, consists of maintaining a constant number of episomal copies. After this initial step, the maintenance phase is initiated. This phase consists of creating the conditions to maintain a constant number of viral genomes in the nuclei of undifferentiated basal cells, to create a persistent infection. E1, E2, E6 and E7 are required for stable episomal maintenance of HPV. Finally, vegetative or productive viral replication is initiated with the subsequent production of progeny virions.

**Figure 2 vaccines-10-00731-f002:**
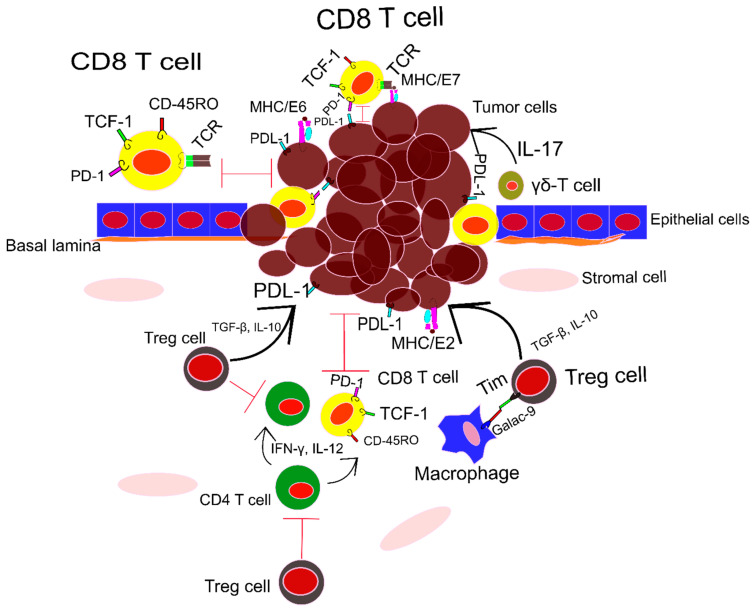
The main subsets of T cells in the HPV-related cancers are CD4 regulatory T cells, CD8 T cells, and γδ T cells. CD8 T cells are mostly tracked in the epithelial layer. CD8 TCRs recognize viral antigens attached to MHC-I on the surface of tumor cells. It helps the eradication of tumor cells. However, tumor cells with PDL-1 receptors on their cell interact with PD-1 on T cells and impair the CD8 T cells. In spite, CD4 T cells infiltrated into the stromal layer. The majority of CD4 T cells in the stromal layer are Treg cells. They promote tumor development by producing IL-10, and TGF-β. These cytokines induce tumor tolerance in other types of T cells. The unconventional γδ T cells are one of the main players in the epithelial layer. It seems these cells promote tumor progression by producing IL-17. For more information read the main text.

**Figure 3 vaccines-10-00731-f003:**
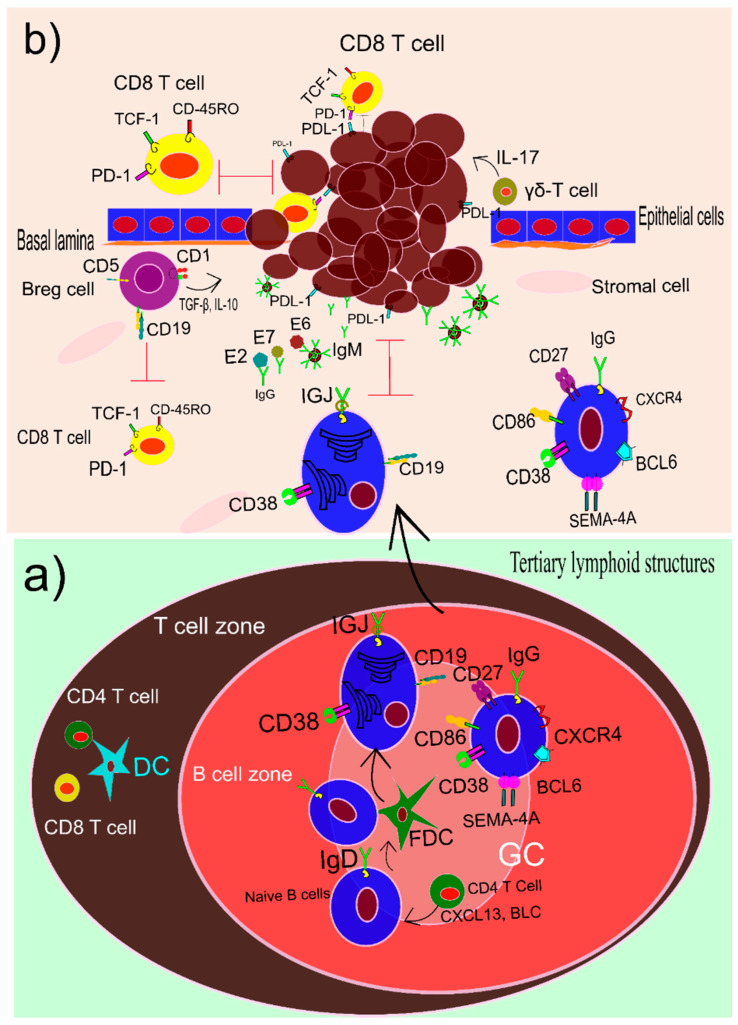
(**a**) The tertiary lymphatic structures (TLSs) appear in the HPV-related tumor tissues due to the chronic infection with HPV. These TLSs are the main player in the maturation of B cells and plasma cells. The cells produce a large number of antibodies against HPV antigens. (**b**) In the HPV-related tumor microenvironment mature B cells, plasma cells, and B regulatory cells (Breg) have a pivotal role. Plasma cells and B cells have a protective contribution and increase the overall survival of patients. In contrast, Breg cells are pro-tumor and support tumor metastasis. Both plasma and B cells secret monoclonal IgG and IgM antibodies against major antigenic proteins of HPV such as E6, E7, and E2. The Breg cells interfere with CD8 T cells and produce cytokines such as IL-10 and TGF-β that promote tumor evasion.

**Figure 4 vaccines-10-00731-f004:**
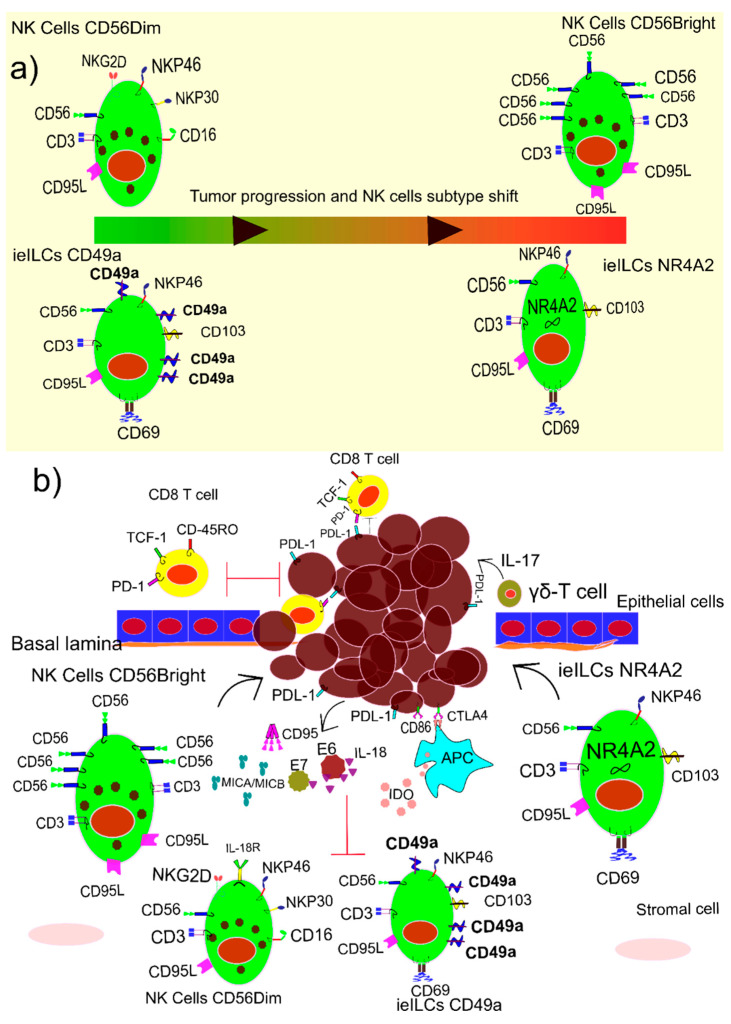
(**a**) The quantity of NK cells in HPV-related cancer does not change. However, there is a shift from subsets of NK cells in the tumor microenvironment that makes the tumor more aggressive and unpredictable. Two classical subtypes of NK cells (CD56 bright and CD56 dim), ieILCs CD49a+, and ieILCs CD49a-/NR4A2+ were studied in the tumor microenvironment. HPV and progressed tumors disrupt the NK cell population balance. The number of NK cells CD56 dim that are preventative effects on tumor development is attenuated, while NK cells CD56 bright are increased. (**b**) The rise in CD56 bright cells is in parallel with the higher frequency of lesions. In the case of ieILCs, CD49A+ has cytotoxic effects on the tumor cells. In contrast, ieILCs CD49a-/NR4A2+ induces tumor progression. In addition, HPV-E6 and E7 disrupt the IFN-γ production by NK cells by withdrawing IL-18 from the tumor environment. IL-18 attachment to its receptor stimulates intracellular cascades and consequently IFN-γ production. Furthermore, abnormal expression of CTLA-4 e on the tumor cells interacts with CD86 on the antigen-presenting cells (APCs). The CTLA-4/CD86 interaction stimulate the APCs for IDO production. IDO impairs the NK cells’ cytotoxicity effects.

**Figure 5 vaccines-10-00731-f005:**
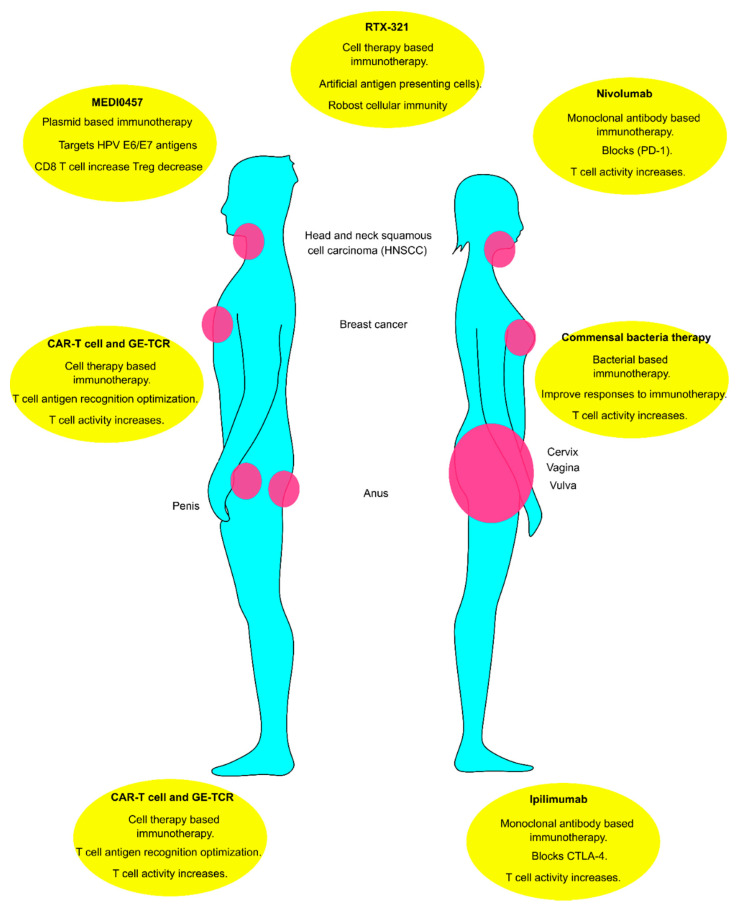
Immunotherapy of HPV-related tumors can be classified into monoclonal antibodies therapy, cell therapy, engineered nucleic acid-based therapies, and bacterial or engineered bacterial therapy. Monoclonal antibodies such as ipilimumab and Nivolumab that target CTLA-4 and PD-1 have been commercially available for a longer time. Recent development in cell therapy and genetic engineering opened a new window for immunotherapy of cancer, especially HPV-related tumors. Manipulation of T cell receptors (TCRs) and chimeric antigen receptor T cells (CAR-T) have been employed for a variety of phases in clinical trials of HPV-related tumors. Indirect cell therapy using plasmid, MEDI0457, to increase and optimize the T cells cytotoxicity shows promising results recently. The majority of the patients show resistance to immunotherapy by CAR-T cells due to mutations in their HLA-A system. Employing artificial antigen receptors such as RTX-321 helps to overcome the limitation of CAR-T cell therapies. Furthermore, bacterial studies of HPV-related tumors demonstrate the significance of residential bacteria in the efficacy of immune responses or HPV evasion. So, bacterial or engineered bacteria have a high potential for indirectly and directly impacting the immunotherapy of HPV-positive tumors.

**Table 1 vaccines-10-00731-t001:** HPV types and associated lesions.

HPV Neoplasia Potential	High-Risk (HR)	Low-Risk (LR)
**HPV Types**	16, 18, 23, 31, 33, 35, 39, 45, 51, 52, 53, 56, 58, 66, 68, 73, and 82	40, 42, 43, 44, 53, 54, 61, 72, 73, and 81
**Lesions**	Intraepithelial neoplasia and cervical cancer	Intraepithelial neoplasia, genital or cutaneus warts

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
