# Peer review of "Immunological Aspects of Human Papilloma Virus-Related Cancers Always Says, “I Am like a Box of Complexity, You Never Know What You Are Gonna Get”"

_vaccines, 2022, doi:10.3390/vaccines10050731_

Round 1

Reviewer 1 Report

Usually, a manuscript has an introduction, materials and methods, results, discussion and conclusion. This manuscript has only introduction and conclusion.

This paper would benefit from review by a native English speaker to assist with the sentence construction. Many of the sentences make no sense, neither logical nor scientific.

I'm not sure if Soleymaninejadian can distinguish between good and bad publications. It is important knowing how to find and use credible information sources. It is important to have understanding of the subject matter, at least in the context of its intended use and audience. The manuscript should be reviewed by people with a scientific background in HPV, microbiology, immunology and cancer research before resubmission.

I'm not sure if Soleymaninejadian can distinguish between primary and secondary sources. Primary sources provide raw information and first-hand evidence. Examples include interview transcripts, statistical data, and works of art. A primary source gives you direct access to the subject of your research. Secondary sources provide second-hand information and commentary from other researchers.

Secondary sources often are defined in contrast to primary sources. In a primary source, an author shares his or her original research—whether it be case study findings, experiment results, interview materials, or clinical observations. However, in a secondary source, an author focuses on presenting other scholars’ research, such as in a literature review.

When trying to distinguish between a primary and secondary source, it is important to ask yourself:

Who originally made the discoveries or brought the conclusions in this document to light?

Did the author conduct the study his or herself?

Or is the author recounting the work of other authors?

Secondary sources refer to sources that report on the content of other published sources.

Citing a source within a source (citing a secondary source) is generally acceptable within academic writing as long as these citations are kept to a minimum. You should use a secondary source only if you are unable to find or retrieve the original source of information.

Major revisions

Line 2-3, title “Immunological aspects of human papilloma virus related cancers always says: “I am like a box of complexity, you never know what you are gonna get””

This is not a scientific title to be used on a scientific paper. Please use another title. The simple solution is not to publish this manuscript in a scientific journal.

Line 10, abstract, “The human papillomavirus (HPV) is attributed to different cancers in both women and men” => “The human papillomavirus (HPV) can cause different cancers in both men and women”

Line 11, abstract, “The virus interferes with functions of the cervix”

The main problem with cervical cancer is not the alteration of functions of the cervix. The main cause of cervical cancer-related death is distant metastasis to internal organs and brain. Please reformulate.

Line 12, abstract, “The tumors lead to death if not treated”

This is the case for most cancers, not only cancers caused by HPV. Delete this sentence.

Line 14-15, abstract, “The tumors know how difficult is to win the battle with a strong united army of immune cells that are equipped with cytokines and enzymes”

The tumors do not have a brain. The tumors do not think. The tumors know nothing. Please reformulate.

Line 16, abstract, “The majority of the time tumors win the battle without being capable to do it”

This is unclear and confusing. Please reformulate.

Line 27-28, introduction, “Human papillomavirus (HPV) infection is a common and usually sexually transmissible infection(STI) in the world” => “Human papillomavirus (HPV) is the most common sexually transmitted infection (STI) in the world”

https://www.who.int/news-room/fact-sheets/detail/cervical-cancer

Line 28, introduction, “with a high negative impact on individual social life”

Most people with HPV do not know they have the infection. They never develop symptoms or health problems from it. Please reformulate.

https://www.cdc.gov/std/hpv/stdfact-hpv.htm

Line 28, introduction, reference 1 is not a primary source. Please find another reference.

Line 32, introduction, reference 2 is not a primary source. Please find another reference.

Line 32-33, introduction, “HPV types can remain latent and go on to cause cancer years later”

The main problem with HPV-infections and risk of cancer development is not latency, but type-specific persistent infections. Please reformulate. You also need a reference.

Walboomers JM, Jacobs MV, Manos MM, Bosch FX, Kummer JA, Shah KV, Snijders PJ, Peto J, Meijer CJ, Muñoz N. Human papillomavirus is a necessary cause of invasive cervical cancer worldwide. J Pathol. 1999 Sep;189(1):12-9. doi: 10.1002/(SICI)1096-9896(199909)189:1<12::AID-PATH431>3.0.CO;2-F. PMID: 10451482.

https://pubmed.ncbi.nlm.nih.gov/10451482/

Line 33-34, “Globally, the greatest burdenof HPV-related cancers is cervical cancer and was identified in 84% of patients” => “Globally, cervical cancer is the most common of the HPV-related cancers and accounts for 84% of the cases”

Line 36, introduction, reference 4 is not “Roman and collegues” and incidence of head and neck cancer in the US, but a Chinese meta-analysis of HPV-genotypes in CIN. Please find another reference, for example:

Roman BR, Aragones A. Epidemiology and incidence of HPV-related cancers of the head and neck. J Surg Oncol. 2021 Nov;124(6):920-922. doi: 10.1002/jso.26687. Epub 2021 Sep 23. PMID: 34558067; PMCID: PMC8552291.

https://pubmed.ncbi.nlm.nih.gov/34558067/

Lechner, M., Liu, J., Masterson, L. et al. HPV-associated oropharyngeal cancer: epidemiology, molecular biology and clinical management. Nat Rev Clin Oncol (2022). https://doi.org/10.1038/s41571-022-00603-7

https://pubmed.ncbi.nlm.nih.gov/35105976/

Line 50, introduction, reference 6 is not a primary source. I will suggest:

zur Hausen, H. Papillomaviruses and cancer: from basic studies to clinical application. Nat Rev Cancer 2, 342–350 (2002). https://doi.org/10.1038/nrc798

Line 52, introduction, reference 7 is not a primary source. I will suggest:

zur Hausen, H. Papillomaviruses and cancer: from basic studies to clinical application. Nat Rev Cancer 2, 342–350 (2002). https://doi.org/10.1038/nrc798

Line 58-59, introduction, “A subgroup of 12 mucosal HPV 58 (HPV16, 18, 31, 33, 35, 39, 45, 51, 52, 56, 58, and 59) are referred to as high-risk (HR) HPV types”

In table 1 (line 121) you have listed 16 HPV-types as high-risk HPV-types. What is correct 12 or 16 hrHPV types? Please explain.

Line 60-61, introduction, “Eight other HPV types (HPV26, 53, 66, 67, 68, 70, 73, and 82), are classified as low-risk (LR) or non-oncogenic and cause benign lesions”

This is wrong. These eight HPV-types are not low-risk types or causes benign lesions. It has been shown that HPV68, HPV26, HPV66, HPV67, HPV73 and HPV82 were significantly more common in cancer cases than in women with normal cervical cytology (Arbyn 2014).

Arbyn M, Tommasino M, Depuydt C, Dillner J. Are 20 human papillomavirus types causing cervical cancer? J Pathol. 2014 Dec;234(4):431-5. doi: 10.1002/path.4424. PMID: 25124771.

https://pubmed.ncbi.nlm.nih.gov/25124771/

Line 71, introduction, reference 18 is not a primary source. Please find another reference.

Line 73, introduction, reference 19 is not relevant. Please find another reference.

Line 79, introduction, reference 18 is not a primary source. Please find another reference.

Line 87, introduction, reference 19 is not relevant. Please find another reference.

Line 115-116, introduction, “The most cervical HPV infections (>90%) are resolved by the host immune system within 1–2 years without a persistent infection”

What is the definition of a persistent infection?

Line 123, introduction, “T cells play a critical role in HPV-leading cancers” => “T cells play a critical role in HPV-related cancers”

Line 127-128, “A case-control study on Egyptian women shows the importance of the CD4 T and CD8 T cells in predicting the progression of HPV-related cancers such as breast, head-neck carcinoma”

Is breast cancer caused by HPV?

Line 131-133, “In the population with normal CD28+ T cells frequency, HPV-2 and HPV-4 related warts cannot survive. Despite this, skin lesions and warts are seen frequently in CD28- T cells patients”

Despite what?

Line 134, “During HPV infection, subpopulations of T cells are infiltrated into different parts of tissues”

What types of tissue? Which parts? Please explain. HPV infect the cutaneous and mucosal epithelium.

Line 135, “In cervical cancer patients infected with HPV-16, T CD8 cells are predominant. In spite, the frequency of CD4 T cells is decreased”

In spite of what?

Line 149, “It sounds in the samples that are taken from the epithelial layer, the CD8 T cells population is dominant”

What sounds?

Line 150, “In spite, in the stromal layer frequency of CD4 T cells is higher”

In spite of what?

Line 152 and 153, “In the majority of cases, these talks make the tumor survive”

Unclear. Please reformulate.

Line 153-155, “A systematic review and meta-analysis were conducted by Litwin et al. [41] show as the progress of the HPV infected tissue toward cancer, the number of T cell subtypes decreases in the tissue”

Does the number of T-cell decreases or the number of subtypes decreases?

Line 168, “Moreover, the majority of CD8 T cells in the infected area are exhausted”

Are you sure?

Line 171-172, “In addition to the conventional T cells, gamma delta T cells (γδ T cells), especially a

rare population that produce the IL-17 play a key role in the progression of HPV-related breast cancer”

What percentage of breast cancer is caused by HPV?

Line 196-197, “Based on Cox regression statistical method if it decreases more than 7.5 folds the mortality rate will increase more than a dozen months”

Unclear. Please reformulate. Mortality rate, or death rate, is a measure of the number of deaths (in general, or due to a specific cause) in a particular population, scaled to the size of that population, per unit of time. Mortality rate can not “increase more than a dozen months”.

Line 270-271, “The pivotal role of Natural Killer (NK) cells in HPV was reported recently in patients with almost normal in all types of immune cells except mature NK cells” => “The pivotal role of Natural Killer (NK) cells in HPV was recently reported in patients with

normal levels of all types of immune cells, with the exception of mature NK cells”

Line 270-327, 1.6. NK cells and HPV

The whole paragraph is unclear. Please reformulate.

Line 329-330, “Impairment of the immune system as a result of chronic infection of HPV notifies the importance of immunotherapy in this event”

Unclear. Please reformulate.

Line 330-331, “Clinical trials divulge the effectiveness of Immunotherapy for HPV-positive tumors” => “Clinical trials have documented the effect of immunotherapy in patients with HPV-positive tumors”

Line 331-332, “In a series, of clinical trials are known as KEYNOTEs, HPV-related tumors were responded optimistically to immunotherapy” => “In a series of clinical trials known as KEYNOTEs, patients with HPV-related tumors responded well to immunotherapy”

Line 333-334, “A systematic review and meta-analysis that eleven studies were included notifies the better overall surveillance of patients” => “A systematic review and meta-analysis that included eleven studies showed better overall survival in patients treated with immunotherapy”

Line 407-409, “Another important aspect of HPV-related cancer is shifting in the commensal bacteria of tissue such as the vagina and oropharynx and their effects on the innate immunity of the patients”

I am not sure about the commensal bacteria and impact of immunity.

Line 413-415, “Furthermore, commensal bacteria improve the overall responses of immunotherapy procedures. The efficacy of anti-PD1 decreases if bacterial flora of the gut is disturbed”

Reference 78 is about metastatic melanoma patients. I am not sure if this is relevant for HPV-related cancers. Reference 76 is from 2014, reference 77 from 2019 and refence 78 from 2018. You can see if Lebenau 2022 have more updated references.

Lebeau, A., Bruyere, D., Roncarati, P. et al. HPV infection alters vaginal microbiome through down-regulating host mucosal innate peptides used by Lactobacilli as amino acid sources. Nat Commun 13, 1076 (2022). https://doi.org/10.1038/s41467-022-28724-8

Figure 4, “Commensal bacteria therapy. Bacterial based immunotherapy”

I am not sure about this.

Line 428-431, “Furthermore, bacterial studies of HPV-related tumors demonstrate the significance of residential bacteria in the efficacy of immunotherapy. So, bacterial or engineered bacteria have a high potential for indirectly and directly impacting the immunotherapy of HPV-positive tumors”

I am not sure about this.

Line 451-453, “A combination of engineered bacteria or oncolytic viruses with immunotherapy can revolutionize the immunotherapy of HPV-related cancers”

If you have not written anything about engineered bacteria or oncolytic viruses in the result section, you can not include this in the conclusion.

Author Response

Dear Editorial board and Reviewers

We are really grateful for second chance has given to us to revise draft of our paper entitled” Immunological aspects of human papilloma virus related cancers always says: “I am like a box of complexity, you never know what you are gonna get.” and resubmit it to “Vaccines (ISSN 2076-393X)”. We are thankful of all the reviewers’ efforts and times. We appreciate their valuable comments and feedbacks. We have been capable to make most of the changes that have been suggested by reviewers.

Comments from Reviewer 1:

Comment 1:

 “This is not a scientific title to be used on a scientific paper. Please use another title. The simple solution is not to publish this manuscript in a scientific journal.”

Response:

Thank you, have raised an important point here. However in this case, the name of paper is traced back to one the most watched movies “Forrest Gump”.  We are not the first one who got the benefits of movies name to attract more readers. For example “Sex, Lies, and Herbicides” in Nature biotechnology that the title named after “Sex, Lies, and Videotapes”. Some other examples are:

  • The Good, the Bad, and the Ugly
    • “The Good, the Bad, and the Outsourced”
    • “The Good, the Bad, and the Whole Grain
    • “The Good, the Bad, and the Cell Type-Specific Roles of Hypoxia Inducible Factor-1 in Neurons and Astrocytes.”
  • Sex, Lies, and Videotape
    • “Sex, Lies, and Insurance Coverage”
    • “Sex, Flies, and Microassays.”
  • Everything You Wanted to Know About Sex (But Were Afraid to Ask)
    • “Everything you always wanted to know about Amorphophallus, but were afraid to stick your nose into!”
    • “Everything You Always Wanted to Know about Copula Modeling but Were Afraid to Ask”
    • “Everything you always wanted to know about protein kinases but were afraid to ask.”

Comment 2:

Line 10, abstract, “The human papillomavirus (HPV) is attributed to different cancers in both women and men => “The human papillomavirus (HPV) can cause different cancers in both men and women””.

 Response:

Agree, we have changed it according to your suggestion.

Comment 3:

“Line 11, abstract, “The virus interferes with functions of the cervix”. The main problem with cervical cancer is not the alteration of functions of the cervix. The main cause of cervical cancer-related death is distant metastasis to internal organs and brain. Please reformulate.”

Response:

Agree, we have changed it according to your suggestion.

Comment 4:

Line 12, abstract, “The tumors lead to death if not treated”. This is the case for most cancers, not only cancers caused by HPV. Delete this sentence.

Response:

Agree, we have changed it according to your suggestion

Comment 5:

Line 14-15, abstract, “The tumors know how difficult is to win the battle with a strong united army of immune cells that are equipped with cytokines and enzymes”. The tumors do not have a brain. The tumors do not think. The tumors know nothing. Please reformulate.

Response:

You have raised an important point here, however, due to the title that was based on a movie we have tried to make the abstract more interactive and more alive.

Comment 6:

Line 16, abstract, “The majority of the time tumors win the battle without being capable to do it”This is unclear and confusing. Please reformulate

Response:

Agree, we have changed it according to your suggestion

Comment 7:

Line 27-28, introduction, “Human papillomavirus (HPV) infection is a common and usually sexually transmissible infection(STI) in the world” => “Human papillomavirus (HPV) is the most common sexually transmitted infection (STI) in the world”

Response:

Agree, we have changed it according to your suggestion

Comment 8:

Line 28, introduction, “with a high negative impact on individual social life”Most people with HPV do not know they have the infection. They never develop symptoms or health problems from it. Please reformulate.

Response:

Agree, we have changed it according to your suggestion

Comment 9:

Line 28, introduction, reference 1 is not a primary source. Please find another reference.

Response:

Agree, we have changed it according to your suggestion

Comment 10:

Line 32, introduction, reference 2 is not a primary source. Please find another reference.

Response:

Thank you for your suggestion, you are very right. However in this case the primary source “http://www.cdc.gov/std/hpv/hpv-factsheet-march-2014.pdf” has been removed from internet.

Comment 11:

Line 32-33, introduction, “HPV types can remain latent and go on to cause cancer years later”. The main problem with HPV-infections and risk of cancer development is not latency, but type-specific persistent infections. Please reformulate. You also need a reference.

Response:

Agree, we have changed it according to your suggestion

Comment 12:

Line 33-34, “Globally, the greatest burdenof HPV-related cancers is cervical cancer and was identified in 84% of patients” => “Globally, cervical cancer is the most common of the HPV-related cancers and accounts for 84% of the cases”

 Response:

Agree, we have changed it according to your suggestion

Comment 13:

Line 36, introduction, reference 4 is not “Roman and collegues” and incidence of head and neck cancer in the US, but a Chinese meta-analysis of HPV-genotypes in CIN. Please find another reference,

Response:

Agree, we have changed it according to your suggestion

Comment 14:

Line 50, introduction, reference 6 is not a primary source. I will suggest: zur Hausen, H. Papillomaviruses and cancer: from basic studies to clinical application. Nat Rev Cancer 2, 342–350 (2002). https://doi.org/10.1038/nrc798

Response:

Agree, we have changed it according to your suggestion

Comment 15:

Line 52, introduction, reference 7 is not a primary source. I will suggest: zur Hausen, H. Papillomaviruses and cancer: from basic studies to clinical application. Nat Rev Cancer 2, 342–350 (2002). https://doi.org/10.1038/nrc798

Response:

Agree, we have changed it according to your suggestion

Comment 16:

Line 58-59, introduction, “A subgroup of 12 mucosal HPV 58 (HPV16, 18, 31, 33, 35, 39, 45, 51, 52, 56, 58, and 59) are referred to as high-risk (HR) HPV types”In table 1 (line 121) you have listed 16 HPV-types as high-risk HPV-types. What is correct 12 or 16 hrHPV types? Please explain.

Response:

Agree, we have changed it according to your suggestion

Comment 17:

Line 60-61, introduction, “Eight other HPV types (HPV26, 53, 66, 67, 68, 70, 73, and 82), are classified as low-risk (LR) or non-oncogenic and cause benign lesions”This is wrong. These eight HPV-types are not low-risk types or causes benign lesions. It has been shown that HPV68, HPV26, HPV66, HPV67, HPV73 and HPV82 were significantly more common in cancer cases than in women with normal cervical cytology

Response:

Agree, we have changed it according to your suggestion

Comment 18:

Line 71, introduction, reference 18 is not a primary source. Please find another reference.

Response:

Agree, we have changed it according to your suggestion

Comment 19:

Line 73, introduction, reference 19 is not relevant. Please find another reference.

Response:

Agree, we have changed it according to your suggestion

Comment 20:

Line 79, introduction, reference 18 is not a primary source. Please find another reference.

Response:

Agree, we have changed it according to your suggestion

Comment 21:

Line 87, introduction, reference 19 is not a primary source. Please find another reference.

Response:

Agree, we have changed it according to your suggestion

Comment 22:

Line 115-116, introduction, “The most cervical HPV infections (>90%) are resolved by the host immune system within 1–2 years without a persistent infection”What is the definition of a persistent infection?

Response:

Agree, we have changed it according to your suggestion

Comment 23:

Line 123, introduction, “T cells play a critical role in HPV-leading cancers” => “T cells play a critical role in HPV-related cancers”

Response:

Agree, we have changed it according to your suggestion

Comment 24:

Line 127-128, “A case-control study on Egyptian women shows the importance of the CD4 T and CD8 T cells in predicting the progression of HPV-related cancers such as breast, head-neck carcinoma”Is breast cancer caused by HPV?

Response:

You have raised an important point here. However recent publication shows high risk HPV, especially HPV-16 in breast cancer. In addition to the reference we have given, number 37, for more information please read the following references:

Li J, Ding J, Zhai K. Detection of Human Papillomavirus DNA in Patients with Breast Tumor in China. PLoS One 2015;10:e0136050. 10.1371/journal.pone.0136050

de Villiers EM, Sandstrom RE, zur Hausen H, et al. Presence of papillomavirus sequences in condylomatous lesions of the mamillae and in invasive carcinoma of the breast. Breast Cancer Res 2005;7:R1-11. 10.1186/bcr940

Wang T, Chang P, Wang L, et al. The role of human papillomavirus infection in breast cancer. Med Oncol 2012;29:48-55. 10.1007/s12032-010-9812-9

Comment 24:

Line 131-133, “In the population with normal CD28+ T cells frequency, HPV-2 and HPV-4 related warts cannot survive. Despite this, skin lesions and warts are seen frequently in CD28- T cells patients”Despite what?

Response:

Agree, we have changed it according to your suggestion

Comment 25:

Line 134, “During HPV infection, subpopulations of T cells are infiltrated into different parts of tissues” What types of tissue? Which parts? Please explain. HPV infect the cutaneous and mucosal epithelium.

Response:

Agree, we have changed it according to your suggestion

Comment 26:

Line 135, “In cervical cancer patients infected with HPV-16, T CD8 cells are predominant. In spite, the frequency of CD4 T cells is decreased”In spite of what?

Response:

Agree, we have changed it according to your suggestion

Comment 27:

Line 149, “It sounds in the samples that are taken from the epithelial layer, the CD8 T cells population is dominant”What sounds?

 Response:

Agree, we have changed it according to your suggestion

Comment 28:

Line 150, “In spite, in the stromal layer frequency of CD4 T cells is higher”In spite of what?

Response:

Agree, we have changed it according to your suggestion

Comment 29:

Line 152 and 153, “In the majority of cases, these talks make the tumor survive”Unclear. Please reformulate.

Response:

Agree, we have changed it according to your suggestion

Comment 30:

Line 153-155, “A systematic review and meta-analysis were conducted by Litwin et al. [41] show as the progress of the HPV infected tissue toward cancer, the number of T cell subtypes decreases in the tissue”Does the number of T-cell decreases or the number of subtypes decreases?

Response:

Thank you for your comment ,“the number of T cell subtypes decreases”

Comment 31:

Line 168, “Moreover, the majority of CD8 T cells in the infected area are exhausted”Are you sure?

Response:

Thank you for your comment. For more information please read reference number 36:

Krishna, S.; Ulrich, P.; Wilson, E.; Parikh, F.; Narang, P.; Yang, S.; Read, A.K.; Kim-Schulze, S.; Park, J.G.; Posner, M.; et al. Human Papilloma Virus Specific Immunogenicity and Dysfunction of CD8 + T Cells in Head and Neck Cancer. Cancer Res. 2018, 78, 6159–6170, doi:10.1158/0008-5472.CAN-18-0163.

Comment 32:

Line 171-172, “In addition to the conventional T cells, gamma delta T cells (γδ T cells), especially a rare population that produce the IL-17 play a key role in the progression of HPV-related breast cancer”What percentage of breast cancer is caused by HPV?

Response:

  • Thank you for your comment. Based on a meta-analysis has done by Li et al. 2010, more than 24% of breast cancers were related to HPV. For more information please read: 1007/s10549-010-1128-0 ,https://doi.org/10.1186/s12885-019-5286-0,

Comment 33:

Line 196-197, “Based on Cox regression statistical method if it decreases more than 7.5 folds the mortality rate will increase more than a dozen months”Unclear. Please reformulate. Mortality rate, or death rate, is a measure of the number of deaths (in general, or due to a specific cause) in a particular population, scaled to the size of that population, per unit of time. Mortality rate can not “increase more than a dozen months”.

Response:

 Agree, we have changed it according to your suggestion

Comment 34:

Line 270-271, “The pivotal role of Natural Killer (NK) cells in HPV was reported recently in patients with almost normal in all types of immune cells except mature NK cells” => “The pivotal role of Natural Killer (NK) cells in HPV was recently reported in patients with normal levels of all types of immune cells, with the exception of mature NK cells”

Response:

Agree, we have changed it according to your suggestion

Comment 35:

Line 270-327, 1.6. NK cells and HPV. The whole paragraph is unclear. Please reformulate.

 Response:

Agree, we have changed it according to your suggestion

Comment 36:

Line 329-330, “Impairment of the immune system as a result of chronic infection of HPV notifies the importance of immunotherapy in this event”.Unclear. Please reformulate.

Response:

Agree, we have changed it according to your suggestion

Comment 37:

Line 330-331, “Clinical trials divulge the effectiveness of Immunotherapy for HPV-positive tumors” => “Clinical trials have documented the effect of immunotherapy in patients with HPV-positive tumors

Response:

Agree, we have changed it according to your suggestion

Comment 38:

Line 331-332, “In a series, of clinical trials are known as KEYNOTEs, HPV-related tumors were responded optimistically to immunotherapy” => “In a series of clinical trials known as KEYNOTEs, patients with HPV-related tumors responded well to immunotherapy”

Response:

Agree, we have changed it according to your suggestion

Comment 39:

Line 333-334, “A systematic review and meta-analysis that eleven studies were included notifies the better overall surveillance of patients” => “A systematic review and meta-analysis that included eleven studies showed better overall survival in patients treated with immunotherapy”

Response:

Agree, we have changed it according to your suggestion

Comment 40:

Line 407-409, “Another important aspect of HPV-related cancer is shifting in the commensal bacteria of tissue such as the vagina and oropharynx and their effects on the innate immunity of the patients. I am not sure about the commensal bacteria and impact of immunity.

 Response:

Agree, we have changed it according to your suggestion. For more information, you can read these references.

Lebeau, A., Bruyere, D., Roncarati, P. et al. HPV infection alters vaginal microbiome through down-regulating host mucosal innate peptides used by Lactobacilli as amino acid sources. Nat Commun 13, 1076 (2022). https://doi.org/10.1038/s41467-022-28724-8

Onywera, H., Williamson, A. L., Mbulawa, Z. Z. A., Coetzee, D., and Meiring, T. L. (2019). The cervical microbiota in reproductive-age South African women with and without human papillomavirus infection. Papillomavirus Res. 7:154–163. doi: 10.1016/j.pvr.2019.04.006

Godoy-Vitorino, F., Romaguera, J., Zhao, C., Vargas-Robles, D., Ortiz-Morales, G., Vazquez-Sanchez, F., et al. (2018). Cervicovaginal fungi and bacteria associated with cervical intraepithelial neoplasia and high-risk human papillomavirus infections in a hispanic population. Front. Microbiol. 9:2533. doi: 10.3389/fmicb.2018.02533

Huang, X., Li, C., Li, F., Zhao, J., Wan, X., and Wang, K. (2018). Cervicovaginal microbiota composition correlates with the acquisition of high-risk human papillomavirus types. Int. J. Cancer 143, 621–634. doi: 10.1002/ijc.31342

Klein, C., Gonzalez, D., Samwel, K., Kahesa, C., Mwaiselage, J., Aluthge, N., et al. (2019). Relationship between the cervical microbiome, HIV status, and precancerous lesions. mBio 10:e02785-18. doi: 10.1128/mBio.02785-18

Comment 41:

Line 413-415, “Furthermore, commensal bacteria improve the overall responses of immunotherapy procedures. The efficacy of anti-PD1 decreases if bacterial flora of the gut is disturbed”Reference 78 is about metastatic melanoma patients. I am not sure if this is relevant for HPV-related cancers. Reference 76 is from 2014, reference 77 from 2019 and refence 78 from 2018. You can see if Lebenau 2022 have more updated references.

Response:

Agree, we have changed it according to your suggestion. Excuse us for not being clear, however, here we didn’t want to show the relevancy of HPV and cancer. We just want to show the importance of microbiata in improving the immunotherapy techniques.

Comment 42:

Figure 4, “Commensal bacteria therapy. Bacterial based immunotherapy” I am not sure about this.

Response:

Agree, we have changed it according to your suggestion

Comment 43:

Line 428-431, “Furthermore, bacterial studies of HPV-related tumors demonstrate the significance of residential bacteria in the efficacy of immunotherapy. So, bacterial or engineered bacteria have a high potential for indirectly and directly impacting the immunotherapy of HPV-positive tumors”I am not sure about this.

Response:

Agree, we have changed it according to your suggestion.

Comment 44:

Line 451-453, “A combination of engineered bacteria or oncolytic viruses with immunotherapy can revolutionize the immunotherapy of HPV-related cancers”If you have not written anything about engineered bacteria or oncolytic viruses in the result section, you can not include this in the conclusion.

Response:

Agree, we have changed it according to your suggestion. Thank you for your suggestion, this part is conclusion and we can give direction to the future research or unanswered questions.

In addition to the above comments, all spelling and grammatical errors pointed out by the reviewers have been corrected. Also, in the T cell section, we add a reference that may help the readers.

We look forward to hearing from you in due time regarding our submission and to responding to any further questions and comments you may have. Sincerely,

Sincerely yours,

Ehsan Soleymaninejadian

04/29/2022

Reviewer 2 Report

Manuscript ID vaccines-1674535. The review manuscript entitled “Immunological aspects of human papilloma virus related cancers always says: “I am like a box of complexity, you never know what you are gonna get” by Dr. Soleymaninejadian provides a detailed overview of the role of oncogenic HPV infection, HPV cycle and the role of the immune system during HPV-driven cancer initiation and progression. The manuscript is potentially interesting as being highly informative and detailed. Despite giving a comprehensive overview on the relationship between oncogenic HPV infection and the anti-tumor/viral immune system, the ms present several inaccurateness. The scientific sound of the abstract is quote often poor. About the main text, several improvements should be made, such as (i) the removing of a large variety of both typo and grammar errors, (ii) the rephrasing of several sentences which are difficult to read, for making the review suitable for publication. Also, figures requiring a serious quality improvement. I therefore recommend a major revision. I have suggestions for improving the manuscript: 

Thank you for letting me revise this manuscript

General comments
1.    A sentence at the end of the introductive section describing the main aims of the review would be helpful for the reader
2.    HPV gene expression and particularly the long control region (LCR) are under epigenetic regulation by DNA methylation (PMID: 21479449 and PMID: 19819061). This information should be detailed 
3.    The quality of all figures should be improved. Several words from the figures should be enlarged as being almost unreadable
4.    When clinical trials are quoted, for example line 344, the clinical trial.gov ID should be included

Minor
Line 10 “is attributed” I would say “is considered the causative factor of..”
Line 28 “nfection(STI)” please include a space between words
Line 38 HPVs belongs to the papillomaviridae family
Line 62 besides cervical cancer, HR-HPVs play a key role during cervical intraepithelial neoplasia (CIN) initiation and progression (DOI: 10.3389/fmicb.2020.591452 and DOI: 10.3389/fonc.2019.00976) This information and supporting references should be included
Lines 70-71 English should be improved
Line 83 for “a” long
Line 113 Please include supporting references for the listed, non-cervical, HPV-driven tumors
Line 128 …such as breast “and” head-neck…
Line 158 “HPV-18, and [35].” Please remove “and”
Line 161 a comma should be included following “interestingly”
Lines 186-196 Please include supporting references 
Lines 212, 229 and 329 This reviewer is not sure about the meaning of “notifies” in these sentences
Line 216 “that B cells”
Line 225 the ref 50 should be also included at the and of this sentence
Line 248 the FIG acronym should be moved under the parenthesis
Line 278 by rhinoctomy.
Line 347 I suggest removing “were”
Lines 355-358 English should eb improved
Line 418 The comma following the word “into,” should be removed
Line 422 “Is getting”
Line 437-439 Please revise the english

Author Response

Comments from Reviewer 2:

Comment 1:

A sentence at the end of the introductive section describing the main aims of the review would be helpful for the reader

Response:

Agree, we have changed it according to your suggestion.

Comment 2:

HPV gene expression and particularly the long control region (LCR) are under epigenetic regulation by DNA methylation (PMID: 21479449 and PMID: 19819061). This information should be detailed 

Response:

Agree, we have changed it according to your suggestion.

Comment 3:

 The quality of all figures should be improved. Several words from the figures should be enlarged as being almost unreadable.

Response:

Agree, we have changed it according to your suggestion.

Comment 4:

 When clinical trials are quoted, for example line 344, the clinical trial.gov ID should be included

Response:

Agree, we have changed it according to your suggestion.

Comment 5:

Line 10 “is attributed” I would say “is considered the causative factor of..”

Response:

Agree, we have changed it according to your suggestion.

Comment 6:

Line 28 “nfection(STI)” please include a space between words

Response:

Agree, we have changed it according to your suggestion.

Comment 7:

Line 38 HPVs belongs to the papillomaviridae family

Response:

Agree, we have changed it according to your suggestion.

Comment 8:

Line 62 besides cervical cancer, HR-HPVs play a key role during cervical intraepithelial neoplasia (CIN) initiation and progression (DOI: 10.3389/fmicb.2020.591452 and DOI: 10.3389/fonc.2019.00976) This information and supporting references should be included

Response:

Agree, we have changed it according to your suggestion.

Comment 9:

Line 113 Please include supporting references for the listed, non-cervical, HPV-driven tumors.

Response:

Agree, we have changed it according to your suggestion.

Comment 10:

Line 128 …such as breast “and” head-neck…

Response:

Agree, we have changed it according to your suggestion.

Comment 11:

Line 158 “HPV-18, and [35].” Please remove “and”

Response:

Agree, we have changed it according to your suggestion.

Comment 11:

Line 161 a comma should be included following “interestingly”

Response:

Agree, we have changed it according to your suggestion.

Comment 12:

Lines 186-196 Please include supporting references 

Response:

Agree, we have changed it according to your suggestion.

Comment 13:

Lines 212, 229 and 329 This reviewer is not sure about the meaning of “notifies” in these sentences

Response:

Thank you for your suggestion. Notify synonyms are reveal, show, demonstrate.

Comment 14:

Line 216 “that B cells”

Response:

Agree, we have changed it according to your suggestion.

Comment 15:

Line 225 the ref 50 should be also included at the and of this sentence

Comment 16:

Line 248 the FIG acronym should be moved under the parenthesis

Comment 17:

Line 278 by rhinoctomy.

Response:

Agree, we have changed it according to your suggestion.

Comment 18:

Line 347 I suggest removing “were”

Response:

Agree, we have changed it according to your suggestion.

Comment 19:

Lines 355-358 English should eb improved

Response:

Agree, we have changed it according to your suggestion.

Comment 16:

Line 418 The comma following the word “into,” should be removed

Response:

Agree, we have changed it according to your suggestion.

Comment 16:

Line 422 “Is getting”

Response:

Agree, we have changed it according to your suggestion.

Comment 16:

Line 437-439 Please revise the english

Response:

Agree, we have changed it according to your suggestion.

In addition to the above comments, all spelling and grammatical errors pointed out by the reviewers have been corrected. Also, in the T cell section, we add a reference that may help the readers.

We look forward to hearing from you in due time regarding our submission and to responding to any further questions and comments you may have. Sincerely,

Sincerely yours,

Ehsan Soleymaninejadian

04/29/2022

Reviewer 3 Report

Authors reviewed immunological aspects of HPV infection in the context of HPV-related cancers and its treatment. The work is interesting and up-to-date. 

However, there are some issues that need to be addressed:

  • In the introduction, the Authors write about the impact of HPV as an STD on social life. Considering the frequency of infection and the possibility of asymptomatic infection and eradication by the body, please elaborate on this potential impact on social life.
  • "... in the United States 34 and other high-income countries, Roman and collegues observed an increased incidence 35 in head and neck cancer compared to cervical cancer..." - It should be clarified that these are HPV-related cancers, not in general, if that was the intention of the Authors.
  • The high risk HPV given in the table do not agree with those mentioned in the text. I am asking for standardization according to the current knowledge.
  • The work requires technical processing, there are many editorial errors, such as no spaces between words, word repetition, incorrect order of figures, eg Figure 2 is first. Also in figures, part "b" appears first.
  • As for the work on HPV immunology and related cancers, there is no reference to vaccination, HPV infection and the risk of cancer growth in people with primary or secondary immunodeficiency.

Author Response

Comments from Reviewer 3:

Comment 1:

In the introduction, the Authors write about the impact of HPV as an STD on social life. Considering the frequency of infection and the possibility of asymptomatic infection and eradication by the body, please elaborate on this potential impact on social life.

Response:

Agree, we have changed it according to your suggestion.

Comment 2:

"... in the United States 34 and other high-income countries, Roman and collegues observed an increased incidence 35 in head and neck cancer compared to cervical cancer..." - It should be clarified that these are HPV-related cancers, not in general, if that was the intention of the Authors

Response:

Agree, we have changed it according to your suggestion.

Comment 3:

The high risk HPV given in the table do not agree with those mentioned in the text. I am asking for standardization according to the current knowledge.

Response:

Agree, we have changed it according to your suggestion.

Comment 2:

The work requires technical processing, there are many editorial errors, such as no spaces between words, word repetition, incorrect order of figures, eg Figure 2 is first. Also in figures, part "b" appears first.

Response:

Agree, we have changed it according to your suggestion.

Comment 2:

As for the work on HPV immunology and related cancers, there is no reference to vaccination, HPV infection and the risk of cancer growth in people with primary or secondary immunodeficiency.

Response:

Agree, we have changed it according to your suggestion.

In addition to the above comments, all spelling and grammatical errors pointed out by the reviewers have been corrected. Also, in the T cell section, we add a reference that may help the readers.

We look forward to hearing from you in due time regarding our submission and to responding to any further questions and comments you may have. Sincerely,

Sincerely yours,

Ehsan Soleymaninejadian

04/29/2022

Round 2

Reviewer 1 Report

The authors have addressed my comments and I am satisfied with the improvements.

Reviewer 2 Report

The ms can be accepted in the preent form

Reviewer 3 Report

No further comments. All previous reviewer comments were addressed.